# Measurement of Urinary Gc-Globulin by a Fluorescence ELISA Technique: Method Validation and Clinical Evaluation in Septic Patients—A Pilot Study

**DOI:** 10.3390/molecules28196864

**Published:** 2023-09-29

**Authors:** Tamás Kőszegi, Zoltán Horváth-Szalai, Dániel Ragán, Brigitta Kósa, Balázs Szirmay, Csilla Kurdi, Gábor L. Kovács, Diána Mühl

**Affiliations:** 1Department of Laboratory Medicine, Medical School, University of Pécs, 7624 Pécs, Hungary; 2János Szentágothai Research Center, University of Pécs, 7624 Pécs, Hungary; 3Hungarian National Laboratory on Reproduction, University of Pécs, 7624 Pécs, Hungary; 4Department of Anesthesiology and Intensive Therapy, Medical School, University of Pécs, 7624 Pécs, Hungary

**Keywords:** urinary Gc-globulin, fluorescence ELISA, sepsis, acute kidney injury, predictive value

## Abstract

A major complication of sepsis is the development of acute kidney injury (AKI). In case of acute tubular damage, Gc-globulin, a known serum sepsis marker is increasingly filtrated into the urine therefore, urinary Gc-globulin (u-Gc) levels may predict septic AKI. We developed and validated a competitive fluorescence ELISA method for u-Gc measurement. Serum and urine samples from septic patients were collected in three consecutive days (T1, T2, T3) and data were compared to controls. Intra- and interassay imprecisions were CV < 14% and CV < 20%, respectively, with a recovery close to 100%. Controls and septic patients differed (*p* < 0.001) in their u-Gc/u-creatinine levels at admission (T1, median: 0.51 vs. 79.1 µg/mmol), T2 (median: 0.51 vs. 57.8 µg/mmol) and T3 (median: 0.51 vs. 55.6 µg/mmol). Septic patients with AKI expressed higher u-Gc/u-creatinine values than those without AKI at T1 (median: 23.6 vs. 136.5 µg/mmol, *p* < 0.01) and T3 (median: 34.4 vs. 75.8 µg/mmol, *p* < 0.05). AKI-2 stage patients exhibited more increased u-Gc/u-creatinine levels at T1 (median: 207.1 vs. 53.3 µg/mmol, *p* < 0.05) than AKI-1 stage individuals. Moderate correlations (*p* < 0.001) were observed between u-Gc/u-creatinine and se-urea, se-creatinine, se-hsCRP, WBC, u-total protein, u-albumin, u-orosomucoid/u-creatinine, and u-Cystatin C/u-creatinine levels. U-Gc testing may have a predictive value for AKI in septic patients.

## 1. Introduction

Sepsis is still a highly challenging and complex syndrome for intensive care patients worldwide. In spite of the development of diagnostic and treatment modalities, the incidence of sepsis is higher than 500 cases per 100,000 person years with a continuously rising trend affecting about 50 million people with a mortality rate between 30–50% [1]. In the last decade, the term sepsis has been thought to be a clinical syndrome rather than a disease, sometimes with only mild early symptoms resulting in a delayed diagnosis [2,3]. Therefore, the definition of sepsis has changed currently to Sepsis-3, with a major focus on life-threatening organ dysfunction due to a dysregulated host response to infection [4,5]. Sepsis-3 definition emphasizes the significance of organ dysfunction, which is defined as two points or more in the Sequential Organ Failure Assessment (SOFA) score.

A major complication of sepsis, and especially septic shock, is the development of acute kidney injury (AKI), quite often requiring renal replacement therapy (RRT) [6]. According to the consensus statement of the 28th Acute Disease Quality Initiative workgroup, sepsis-associated AKI is defined by the presence of both Sepsis-3 criteria [4] and AKI criteria (as defined by Kidney Disease: Improving Global Outcomes recommendations) [7] when AKI develops within 7 days from diagnosis of sepsis [8]. AKI criteria are fulfilled in case of a patient, when there is an increase in serum creatinine level by ≥0.3 mg/dL (≥26.5 µmol/L) within 48 h, or an increase in serum creatinine level to ≥1.5 times compared with baseline within the previous 7 days, or when urine volume decreases ≤0.5 mL/kg/h for 6 h. AKI itself is a heterogeneous syndrome leading to different clinical phenotypes because multiple mechanisms (e.g., inflammation, microcirculatory and mitochondrial dysfunction, nephrotoxic drugs) contribute to organ injury across the course of sepsis. Besides the clinical symptoms, there is still ongoing research for obtaining laboratory parameters as well, to aid diagnosis and to predict future complications with a special emphasis on AKI [9,10].

Unfortunately, currently, there are no gold standard lab parameters with high sensitivity and specificity for diagnosing and/or follow-up of sepsis, and without appropriate tools for predicting outcomes [11]. Up to now, regarding sepsis in general, the most widely applied routine laboratory parameters are high sensitivity C-reactive protein (hs-CRP) and procalcitonin (PCT), the latter with higher specificity than hs-CRP [12]. However, many other markers have been tested for the diagnosis and follow-up of sepsis such as presepsin, calprotectin, proinflammatory interleukins, monocyte chemoattractant protein-1, pentraxin-3, mid-regional fragment of pro-adrenomedullin and many others [13]. Trying to establish novel sepsis markers with a potential predictive value regarding organ dysfunctions as well, during the past few years, our research group has found non-commercially available complementary laboratory tests [10,14,15,16,17,18,19,20,21,22] and adapted most of them to automated laboratory instrumentation including serum gelsolin (GSN), Gc-globulin (Gc), presepsin/GSN ratio, and urinary markers such as orosomucoid (u-ORM), Cystatin C (u-CysC) and neutrophil gelatinase associated lipocalin (u-NGAL).

Several studies have investigated that in case of widespread tissue damage, an excessive amount of cellular (globular [G] and filamentous [F]) actin releases into the circulation, leading to the formation of long filamentous (F) chains that might lead to microcirculatory dysfunction [23,24,25,26]. In addition, high amount of free extracellular G-actin impairs DNase-1–mediated clearance of cell-free (cf) DNA, resulting in persistently elevated blood levels of cytotoxic cfDNA, which contributes to multiple organ dysfunction syndrome [27,28]. Excess actin in the plasma reduces the level of scavenger actin-binding proteins (GSN, Gc-globulin, thymosin beta-4), and excess actin can also be found in the urine of the patients who could have a diagnostic value regarding sepsis-related AKI [10,16,20,21,29,30,31]. Serum Gc-globulin is a low molecular weight (52–59 kDa) glycosylated α2-globulin produced by the liver. The protein owns three major isoforms (Gc1f, Gc1s, Gc2). Besides its monomer actin-binding capacity, Gc-globulin is the major transporter of vitamin D metabolites (therefore its alternative name: vitamin D-binding protein [VDBP]); in addition, it can bind fatty acids and might have a role in modulating cells of the adaptive immune system [32]. Reduced se-Gc-globulin levels have been reported in trauma, acute liver failure and sepsis, where it was found as a marker of organ dysfunctions and mortality [20,31,33]. Because of its low molecular weight, serum Gc-globulin is filtrated through the glomeruli with partial reabsorption and even in healthy individuals, a low amount can be detected within urine [32]. Urinary Gc-globulin has already been investigated in conditions as contrast-induced and acute heart failure-associated kidney injury, diabetic nephropathy, lupus nephritis, nephrosis syndrome, chronic kidney disease, and in gynecological diseases as well [34,35,36,37,38,39,40,41,42,43,44,45,46,47,48]. However, there are only scarce data regarding u-Gc-globulin levels in sepsis. In our study, we postulated that the increased general permeability in sepsis, especially in kidney injury, causes the urinary Gc-globulin levels to increase predicting a potential development of AKI. Therefore, we developed and validated a fluorescence ELISA method for u-Gc determination, and clinically evaluated its diagnostic potential in septic patients with and without AKI.

## 2. Results

### 2.1. Optimization of Antigen Coating and Antibody Dilutions

We used commercial Human Serum Protein Calibrator and Human Serum Protein Low Control diluted to the concentrations found in the urine of healthy individuals and patients with various diseases [34,36,37,38,39,40,42,43,44,45,47]. For optimization of the competitive (inhibition) ELISA, the checkerboard method gave the best results at 31.25 ng/mL final Gc calibrator level for coating of the binding plates, which means 3.125 ng/well Gc-globulin in the matrix of the calibrator. The highest signal was obtained when both the primary and secondary antibodies were diluted to 10,000-fold.

### 2.2. Validation of the Method

The determined LoB, LoD and LoQ values were 2.20, 3.90 and 6.79 ng/mL, respectively. A cumulative calibration curve obtained from 12 independent calibrations (performed totally manually with 96 replicates at each point) is shown in Figure 1A. The LoQ and linearity check are shown in Figure 1B. The recovery in the range of 6.25–90 ng/mL was between 105–108% at each level.

The intra- and inter-assay reproducibility data are shown in Table 1.

### 2.3. Patients’ Demographic and Laboratory Data

In the present study, control and septic patients were enrolled. Basic demographic and admission laboratory data of the 23 control individuals, 13 septic and 28 sepsis-related AKI patients are shown in Table 2. A moderate difference was found between control and septic patient groups regarding age and some comorbidities. A significant difference (*p* < 0.05) was observed between control, septic and sepsis-related AKI groups in se-TP, u-Gc and u-Gc/u-creatinine levels. Both septic patients’ groups exhibited lower (*p* < 0.05) admission se-ALB, se-GSN, se-Gc levels, and higher (*p* < 0.05) white blood cell count, higher hs-CRP, u-ORM, u-Cystatin C levels and u-ORM/u-creatinine, u-Cystatin C/u-creatinine ratios than the control group. Sepsis-related AKI patients had higher u-TP levels than controls. Admission se-creatinine and u-ALB levels were higher (*p* < 0.05) in the sepsis-related AKI group than in the control group; moreover, they also tended to be higher (*p* < 0.05) in sepsis-related AKI patients when compared to septic patients. PCT levels were generally higher in sepsis-related AKI patients than in septic patients, although this increase was not statistically significant (*p* = 0.06).

### 2.4. Septic Patients’ Clinical Data

No significant difference was found between sepsis (n = 13) and sepsis-related AKI (n = 28) groups regarding age, gender, cause of admission, length of ICU stays and 14-day mortality. However, 12 patients required renal replacement therapy (RRT) from the sepsis-related AKI group. Regarding the clinical prediction scores, sepsis-related AKI patients had considerably higher (*p* < 0.05) APACHE II and SAPS II scores compared with the sepsis group, although this increase in SOFA score was not statistically significant (*p* = 0.08). Multiple organ dysfunction syndrome (MODS) was also a more common complication in the sepsis-related AKI group (64.3%) than in the sepsis group (46.2%). The most frequent organ dysfunctions in the sepsis-related AKI and septic patient groups were sepsis-induced hypotension (78.6% vs. 76.9%), followed by acute lung injury (53.6% vs. 38.5%) and thrombocytopenia (21.4% vs. 15.4%). Additional clinical data of septic patients are presented in Table 3.

### 2.5. U-Gc/u-Creatinine Levels in Control, Septic and Sepsis-Related AKI Patients

A considerable difference was found in u-Gc/u-creatinine levels between the control and septic patient groups at T1 (median: 0.51 vs. 79.1 µg/mmol, *p* < 0.001), T2 (median: 0.51 vs. 57.8 µg/mmol, *p* < 0.001) and T3 (median: 0.51 vs. 55.6 µg/mmol, *p* < 0.001), yet there was no significant change regarding the kinetics of u-Gc/u-creatinine levels during the follow-up (T1, T2, T3) of septic patients (Figure 2).

Septic AKI patients exhibited higher u-Gc/u-creatinine levels than septic patients at T1 (median: 136.5 vs. 23.6 µg/mmol, *p* < 0.01) and T3 (median: 75.8 vs. 34.4 µg/mmol, *p* < 0.05) (Figure 3A). AKI-2 stage septic patients showed higher u-Gc-globulin/u-creatinine levels than AKI-1 stage septic individuals at T1 (median: 207.1 vs. 53.3 µg/mmol, *p* < 0.05); in addition, AKI-3 stage septic patients had higher u-Gc-globulin/u-creatinine levels than AKI-1 stage septic patients at T3 (median: 85.4 vs. 17.6 µg/mmol, *p* < 0.05) (Figure 3B).

### 2.6. Survival Data and Distinctive Power of u-Gc Globulin/u-Creatinine Levels in Sepsis

No significant difference was found in u-Gc globulin/u-creatinine levels between survivors and non-survivors regarding 14-day mortality during follow-up (Figure 4A). The diagnostic performance of first-day parameters in sepsis-related AKI was assessed using ROC analysis (Figure 4B). For distinguishing all sepsis-related AKI patients from septic patients without AKI, area under the curve (AUC) values were found to be the following: u-Gc/u-creatinine: 0.772 (95% confidence interval [CI]: 0.629–0.915; *p* < 0.01); se-creatinine: 0.880 (95% CI: 0.770–0.991); *p* < 0.001). The derived cut-off values were for u-Gc/u-creatinine 71.2 µg/mmol (sensitivity: 67.9%, specificity: 84.6%); and for se-creatinine: 139.5 µmol/L (sensitivity: 85.7%, specificity: 84.6%). In our small cohort, u-ORM/u-creatinine and u-CysC/u-creatinine alone did not offer significant diagnostic capacity. However, when we assessed the combined predictive value of the investigated markers, the following promising results were obtained: AUC for the combination of u-Gc/u-creatinine and u-CysC/u-creatinine was 0.777 (95% CI: 0.635–0.920; *p* < 0.01); the combination of u-Gc/u-creatinine and u-ORM/u-creatinine was 0.791 (95% CI: 0.655–0.928; *p* < 0.01),whereas AUC for the combination of u-Gc/u-creatinine, u-ORM/u-creatinine, u-CysC/u-creatinine and se-creatinine was 0.901 (95% CI: 0.809–0.994; *p* < 0.001), respectively.

### 2.7. Correlations

Data from all sample collection time points were used for calculating correlations. U-Gc/u-creatinine levels showed strong correlations (*p* < 0.001) to u-Gc (ρ = 0.923). Moderate correlations (*p* < 0.001) were observed between u-Gc/u-creatinine and se-urea (ρ = 0.399), se-creatinine (ρ = 0.466), se-hs-CRP (ρ = 0.407), WBC (ρ = 0.376), u-TP (ρ = 0.563), u-ALB (ρ = 0.696), u-ORM (ρ = 0.415), u-ORM/u-creatinine (ρ = 0.608), u-Cystatin C (ρ = 0.624) and u-Cystatin C/u-creatinine (ρ = 0.722) levels. U-Gc/u-creatinine ratios negatively correlated (*p* < 0.001) with se-TP (ρ = −0.402), se-albumin (ρ = −0.427), se-GSN (ρ = −0.390) and se-Gc-globulin (ρ = −0.250). No further associations were found with other inflammatory or clinical parameters (Table 4).

## 3. Discussion

Sepsis-related acute kidney injury (AKI) is a life-threatening organ dysfunction developing in 25–75% of the patients during sepsis syndrome with a mortality rate up to 77%. The pathophysiology of sepsis-induced AKI is complex; the interaction between genotype and exposures leads to a variety of clinical phenotypes [8]. For timely diagnosis, treatment and prediction of sepsis-associated AKI, biomarkers measured with rapid assays could be essential laboratory tools. Apart from the gold standard AKI functional marker, se-creatinine, several other markers have been investigated, including AKI stress markers (e.g., TIMP-2 [tissue inhibitor of metalloproteinase-2], IGFBP7 [insulin-like growth factor binding protein]), damage markers (e.g., NGAL, KIM-1 [kidney injury molecule 1]) and functional markers (e.g., Cystatin C, penKID [proenkephalin A 119–159]) with various success [8,10].

In our study we developed and validated a new fluorescent ELISA method for the measurement of u-Gc-globulin levels and investigated u-Gc/u-creatinine among septic patients. Although there are a few commercially available ELISA kits for measuring Gc-globulin from various sample types, these are primarily suggested to use serum and/or plasma samples [34,36,47,48,49,50]. Even if the sensitivity of available ELISA kits would fulfil the requirements for measurement of urinary Gc-globulin, they do not offer significant advantages regarding the incubation times and sensitivity to our proposed method. Our assay could be performed in any laboratories equipped with a fluorescence plate reader at a very low cost.

We found significantly higher first-day u-Gc/u-creatinine levels in septic AKI patients than in septic non-AKI patients and a diagnostic value for u-Gc/u-creatinine regarding sepsis-associated AKI which has not been described yet. During our 3-day follow-up study, after dividing septic patients into non-AKI and AKI groups, significant differences were found in u-Gc/u-creatinine levels at T1 and T3. Moreover, AKI-2 stage septic patients exhibited more increased u-Gc/u-creatinine levels than AKI-1 stage septic individuals at T1, In addition, AKI-3 stage septic patients had higher u-Gc-globulin/u-creatinine levels than AKI-1 stage patients at T3. As far as we know, only two research groups elucidated the predictive role of u-Gc/u-creatinine regarding acute renal dysfunction in different clinical scenarios. Chaykovska et al. [34] found u-Gc-globulin as a promising predictive marker regarding major adverse renal events (dialysis need or doubling of se-creatinine at follow-up, death) in patients undergoing coronary angiography during the 90 days follow-up. However, they did not find u-Gc/u-creatinine as a predictive marker for contrast media-induced nephropathy. In contrast to their observation, we did not note any difference between survivors and non-survivors regarding u-Gc/u-creatinine levels, and we did not find u-Gc/u-creatinine as a marker of mortality. Recently, Diaz-Riera et al. [35] demonstrated increased u-Gc levels in patients suffering from acute decompensated heart failure (ADHF)-associated renal dysfunction at hospital admission when compared to ADHF patients with normal renal function. This difference remained significant also at day 3 after hospital admission, similar to our observations among septic patients with and without AKI. Somewhat in contrast to our results, they did not find u-Gc/u-creatinine alone as a predictive marker regarding ADHF-associated acute renal injury, only when combining it with u-Cystatin C and u-KIM-1. We also tested the combined predictive value of u-Gc/u-creatinine, u-Cys-C/u-creatinine, u-ORM/u-creatinine and the gold standard se-creatinine and found a more robust diagnostic value for sepsis-associated AKI than in case of the individual markers. Our finding underlines the role of the multi-marker approach in AKI, which has already been proposed by several research groups [8,10].

While the results regarding the predictive value of u-Gc/u-creatinine for AKI are scarce, numerous research groups investigated the clinical utility of u-Gc/u-creatinine in diabetic patients. Increased urinary loss of Gc were reported in diabetic patients when compared to controls, which was tightly associated with the grade of albuminuria/nephropathy [36,37,38,39,51]. Moreover, Tian et al. [37] identified u-Gc as a promising diagnostic marker for diabetic nephropathy (DN). Bai et colleagues [39] also concluded that u-Gc combined with serum miR-155-5p, a miRNA highly expressed in DN, had a higher diagnostic value for DN that the single markers alone. In these studies, u-Gc significantly correlated with u-albumin, similarly to our findings in sepsis.

In chronic kidney disease patients, u-Gc significantly correlated with u-total protein [45], and in a recent urinary proteomics study [46], u-Gc was found to be a marker consistently correlating with u-protein and kidney function (se-creatinine, estimated glomerular filtration rate [eGFR]). In a previous, intriguing study, conducted among children on chronic peritoneal dialysis [50], urinary and dialysate Gc excretion closely mirrored albumin losses. The observed correlations between u-Gc and se-creatinine, u-protein and u-albumin are in line with our findings.

U-Gc was also investigated in childhood idiopathic nephrotic syndrome by Bennet et al. [42], where it negatively correlated with eGFR and positively correlated with u-albumin/creatinine, like in our findings. In addition, it had a high discriminatory value regarding steroid-resistant nephrotic syndrome vs. steroid-sensitive nephrotic syndrome [42,43]. Moreover, u-Gc was assessed in childhood-onset systemic lupus erythematosus (SLE) patients diagnosed with lupus nephritis (LN) [40]. Among those young patients, u-Gc alone or in combination with other urinary markers as ORM, lipocalin-like prostaglandin D synthase and transferrin were valuable predictors for a successful response to LN therapy at the age of month 3 or at a later follow-up. Other investigators carried out a study with adult SLE patients and indicated that, among other urinary markers, u-Gc positively correlated with proteinuria and with specific SLE disease activity scores [41].

Based on the literature data, the magnitude of u-Gc/u-creatinine ratios among septic AKI patients in our study tended to be higher than that in diabetic DN patients with microalbuminuria [37], in preeclamptic patients [47] and in patients after contrast media exposure with MARE [34], but lower than in young nephrotic patients [42].

Increased urinary Gc-globulin excretion described in various clinical contexts is suggested to originate from proximal tubular dysfunction of the kidneys. U-Gc is physiologically reabsorbed by multiligand receptors, megalin- and cubilin-mediated endocytosis and catabolized by proximal tubular epithelial cells, which significantly reduces its urinary loss [52]. This mechanism is responsible for scavenging 25(OH)D_3_-Gc-globulin complexes and facilitates the subsequent proximal tubular activation of vitamin D. Apart from some inherited conditions, receptor (megalin and cubilin) dysfunction has been suggested in acquired diseases associated with proteinuria, such as acute kidney injury and chronic kidney disease [52]. This suggested mechanism also strengthens the study of Albejante et al. [47], where preeclamptic patients with proteinuria exhibited large amounts of u-cubilin and Gc excretion when compared to non-proteinuric preeclamptic patients and healthy pregnant women. In our study, we observed positive correlations between u-Gc/u-creatinine and se-creatinine, u-TP, u-ALB, u-ORM/u-creatinine and u-Cystatin C/u-creatinine. In our previous study [15], u-ORM/u-creatinine was found to be a promising diagnostic marker of sepsis and a predictive marker for renal replacement therapy among sepsis-related AKI patients. In another study carried out by our research group [19], u-Cystatin C/u-creatinine was found to be significantly higher in the sepsis-related AKI group than in patients with chronic hypertension and those with type 2 diabetes. Physiologically, both u-ORM and u-Cystatin C are mostly reabsorbed by proximal tubular cells, which is then disturbed in proximal tubular injury. In addition, during the course of sepsis, intrarenal production of ORM is also suggested, however, its role has not been clarified yet [53]. Excess Cystatin C in the urine is described as a marker of proximal tubular injury [54]. The correlations between u-Gc/u-creatinine, u-ORM/u-creatinine and u-Cystatin C/u-creatinine suggest that increased u-Gc excretion in sepsis reflects to proximal tubular injury. However, an increased glomerular permeability may also account for it, as we found a negative correlation between serum and urinary Gc. Recently, a novel marker, urinary actin was suggested to be a promising diagnostic marker for sepsis-associated AKI described also by our research group [21]. In the future, it might be worth investigating the predictive value of both actin and the actin scavenger protein Gc globulin together in sepsis.

Our study is not without limitations. In the future, our results would require validation in larger clinical studies. Urinary Gc-globulin levels were only measured by the new fluorescent ELISA and no data comparison was performed (e.g., Bland–Altman analysis when comparing of our method with commercially available ELISAs). Since the turnaround time of ELISA does not enable it to be implemented in routine laboratory tests in sepsis, our future aim is to develop and validate a rapid, automated particle-enhanced immune turbidimetric method for u-Gc measurement.

## 4. Materials and Methods

### 4.1. Development of a Competitive u-Gc Elisa Assay with Fluorescence Detection

High binding 96-well transparent Elisa plates (Sarstedt, Nümbrecht, Germany) were coated with serum Gc-globulin calibrator (Human Serum Protein Calibrator, ref.: X0-01908-2, DAKO-Agilent, Santa Clara, CA, USA) and diluted with phosphate-buffered saline (PBS, pH 7.2). Into each well, 100 µL of human Gc-globulin calibrator at a final concentration of 31.25 ng/mL regarding Gc in PBS was added. Coating was performed for 1 h at 37 °C in the dark without shaking. After coating, the plates were washed 3 times with 300 µL of T-TBS (Tris buffered saline containing 0.05% Tween 20, pH 7.5). The emptied wells were blocked with 300 µL of SuperBlock T20 (ThermoFisher, Scientific, Waltham, MA, USA), for 3 × 1 min.

For the assay, various dilutions from the calibrators, controls (Human Serum Protein Cal X0-90801-2 and Human Serum Low Prot Control X0-93901-2, DAKO-Agilent, Santa Clara, CA, USA) and samples were prepared, based on previous recommendations [36,49]. The calibrators were diluted in the range of 0–125 ng/mL, while the Human Serum Protein Low Control was in the range of 6.25–125 ng/mL, both in 0.05% Tween 20-TBS (T-TBS). Urine samples were routinely diluted 10-fold with tri-distilled water, but when the obtained u-Gc values exceeded the measuring range, a further 20–40-fold dilutions were used and then retested.

The primary antibody (Polyclonal Rabbit Anti-Human Gc-globulin, code no.: A0021, DAKO-Agilent, Santa Clara, CA, USA) was diluted 10,000-fold with T-TBS. For the competitive assay, the antigens (calibrators, control and samples) were mixed in separate tubes with the antibody in a ratio of 1 to 1. Incubation in the dark was performed for 1 h at 37 °C.

After the incubation period, the blocked wells were emptied and filled with 100 µL volumes of the premixed antigen–antibody complexes. A further incubation of the plates was performed for 1 h at 37 °C in the dark, followed by 3 washes with T-TBS. The secondary antibody (horseradish peroxidase (HRP) labeled Polyclonal Goat Anti-Rabbit IgG, code no.: P0448, DAKO-Agilent, Santa Clara, CA, USA) was diluted 10,000-fold in 1 mg/mL bovine serum albumin (BSA) and dissolved in T-TBS buffer. Into each well, 100 µL of the diluted secondary antibody was added with a further incubation in the dark for 1 h at 37 °C. Then, the plates were washed twice with T-TBS and the final wash was performed with TBS only (300 µL/well).

The immune reaction was developed by the following fluorogenic reagent (AMP): 50 mM K-phosphate buffer of pH 7.4, containing 100 µM H_2_O_2_ dissolved in 0.1% citric acid and 16 µM Ampliflu Red (Merck, Darmstadt, Germany), freshly prepared and kept on ice until usage. In total, 150 µL of AMP was pipetted into each emptied well and, after 5 s of shaking the plates, were then incubated at room temperature in the dark for 60 min. Fluorescence signal was measured by a Perkin Elmer Enspire multimode plate reader (Per-Form Hungaria Ltd., Budapest, Hungary) at 540/580 nm excitation/emission wavelengths. The data in cps vs. calibrator concentrations were evaluated using a non-linear logistic fitting by Origin 2016 program (OriginLab Corporation, Northampton, MA, USA). The u-Gc concentrations of the controls and samples were obtained using the calibration curve’s equation.

A simplified flow chart of our proposed method is shown in Figure 5.

#### Validation of the Method

The validation of our assay was performed by the CLSI guidelines [55,56,57]. For each plate, calibrators were prepared in triplicates. Limit of blank (LoB) was calculated from 4 independent measurements (80 replicates) of blank samples. Limit of detection (LoD) was obtained by using the data of 4 independent measurements of Human Serum Protein Low Controls diluted to a concentration of 2–4 ng/mL (n = 80). For the limit of quantification (LoQ), we used 6 dilutions of a Human Serum Protein Low Control sample from 6.25 to 90 ng/mL concentrations and 7 independent measurements (n = 42 at each point). The LoQ was set at the concentration where the CV reached 15%, taking all u-Gc data into the calculation. Recovery was also calculated from the LoQ data in the whole concentration range studied. The linearity of the method was estimated using linear regression analysis of the Human Serum Protein Low Control dilutions. Intra-assay and inter-assay imprecision were also determined at two concentration levels from 25 replicates (intra-assay) and 80 replicates measured on 10 consecutive days (inter-assay). A cumulative calibration curve was also obtained from 12 independent calibrations (n = 96 replicates at each point).

### 4.2. Patient Enrollment and Study Design

In the present study, 13 septic and 28 sepsis-related AKI patients were enrolled between January 2018 and December 2019 at the Department of Anesthesiology and Intensive Therapy (Medical School, University of Pécs, Hungary) in a follow-up study. Outpatients (n = 23 control individuals) were from the Department of Ophthalmology (Medical School, University of Pécs, Hungary) without acute inflammation (hs-CRP < 5 mg/L), infection or kidney disease. Both controls and patients or their appropriate surrogates were given detailed information regarding our study protocol and a written consent was also obtained from all of them. Defined end points were the withdrawal of consent or death during the study period. Patients were excluded if they were under 18 years of age or where it was not possible to obtain patient consent or consultee approval. The study protocol was authorized by the Regional Research Ethical Committee of the University of Pécs (4327.316-2900/KK15/2011) and was performed according to the ethical guidelines of the 2003 Helsinki Declaration.

Venous blood and urine samples were collected on the same day in the morning. For controls, one sample was obtained, while the septic patients’ specimens were obtained in 3 consecutive days (T1 = on admission, T2 and T3, respectively).

#### Clinical Data and Laboratory Testing

Inclusion criteria for sepsis were the presence of organ dysfunction assessed by Sequential Organ Failure Assessment (SOFA) score (>2), elevated serum procalcitonin (PCT) levels (>2 ng/mL) and suspected/confirmed infection. Patients were regarded to have AKI based on elevated serum creatinine levels (≥1.5—fold increase from the baseline in the last 7 days or ≥26.5 μmol/L increase within 48 h) or with decreased urine output (<0.5 mL/kg/h for 6 h). The diagnosis of sepsis and AKI were established using the Sepsis-3 definitions and Kidney Disease Improving Global Guidelines (KDIGO) guidelines, respectively [4,7]. Acute physiology and chronic health evaluation II (APACHE II) simplified acute physiology score II (SAPS II), SOFA, and multiple organ dysfunction syndrome (MODS) scores were estimated for the first day of intensive care treatment.

Serum total protein (TP), albumin, creatinine, hs-CRP, PCT, gelsolin (GSN), Gc-globulin, blood cell counts were determined by routine automated laboratory methods and/or our published automated tests [18,20] in our accredited Department of Laboratory Medicine, Hungary (NAH 9-0008/2021). Urinary parameters (u-TP, u-albumin, u-orosomucoid, u-Cystatin C) were obtained by our previously published automated techniques [14,19] and u-Gc by the present ELISA method. Urinary values were referred to urinary creatinine as well.

### 4.3. Statistical Analyses

For statistical analysis, IBM SPSS Statistics for Windows, Version 22, and Origin 2016 program (OriginLab Corporation, Northampton, MA, USA) were used. Distribution of continuous variables was evaluated by Shapiro–Wilk test. Because of non-normally distributed data we performed non-parametric tests. For continuous variables Kruskal–Wallis and Mann–Whitney tests, for qualitative variables χ^2^-tests were performed for investigating differences between patient groups. Friedman’s analysis and post hoc Wilcoxon signed-rank tests were executed for follow-up comparisons. Predictive values were assessed by receiver operating characteristic (ROC) curve analysis. Combinatory marker analysis was performed by binary logistic regression followed by ROC analysis. Correlations between quantitative parameters were determined by Spearman’s rank correlation test. Data are expressed as medians and as interquartile ranges (IQR). Changes in the results were considered to be statistically significant at *p* < 0.05.

## 5. Conclusions

Based on our results, urinary Gc-globulin seems to be a promising additional diagnostic marker for sepsis-induced acute kidney injury. U-Gc combined with other glomerular and tubular injury markers (se-creatinine, u-orosomucoid, u-Cystatin-C, u-actin) might provide reliable clinical information regarding sepsis-associated acute kidney injury.

## Figures and Tables

**Figure 1 molecules-28-06864-f001:**
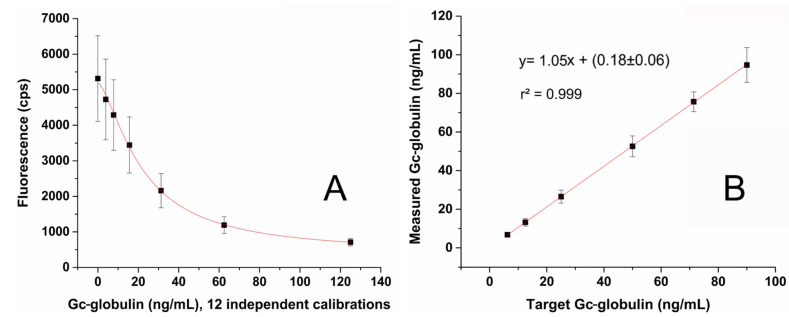
(**A**) Cumulative calibration curve (0–125 ng/mL) for the u-Gc ELISA obtained from 12 independent calibrations (n = 96 replicates/each point) by logistic fitting and (**B**) linearity and recovery of the u-Gc ELISA method calculated from 7 independent tests (n = 42 replicates for each point). Recovery varied from 105 to 108% within the whole concentration range.

**Figure 2 molecules-28-06864-f002:**
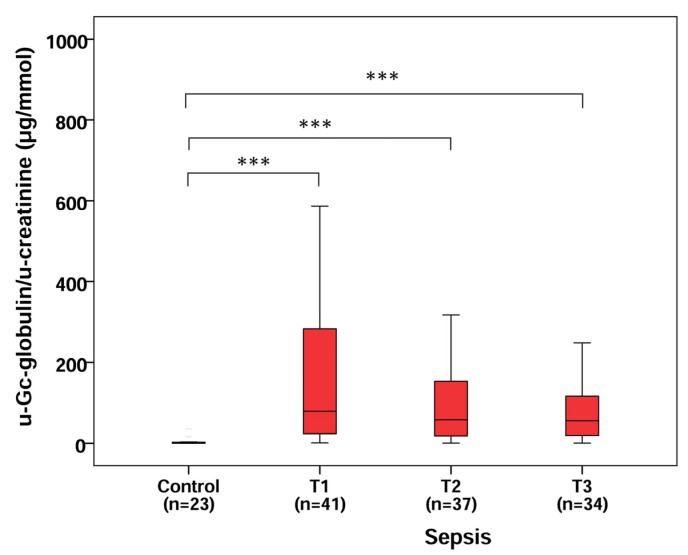
U-Gc-globulin/u-creatinine levels in control and septic patients with a follow-up of septic patients on 3 consecutive days. n: sample count, *** *p* < 0.001.

**Figure 3 molecules-28-06864-f003:**
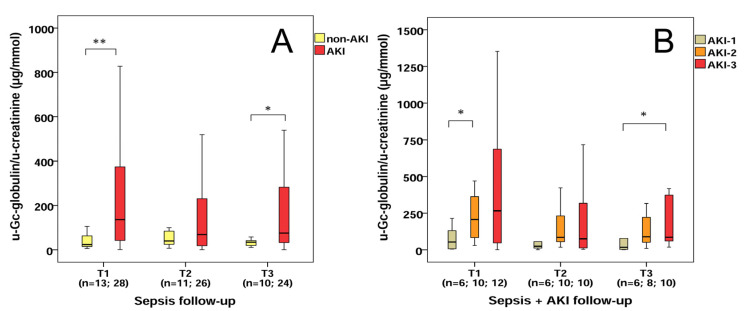
U-Gc/u-creatinine ratios in sepsis-related AKI. U-Gc/u-creatinine levels of septic and sepsis-related AKI patients (**A**) during follow-up. U-Gc/u-creatinine levels regarding sepsis-related AKI stages (**B**) during follow-up. n: sample count, ** *p* < 0.01; * *p* < 0.05.

**Figure 4 molecules-28-06864-f004:**
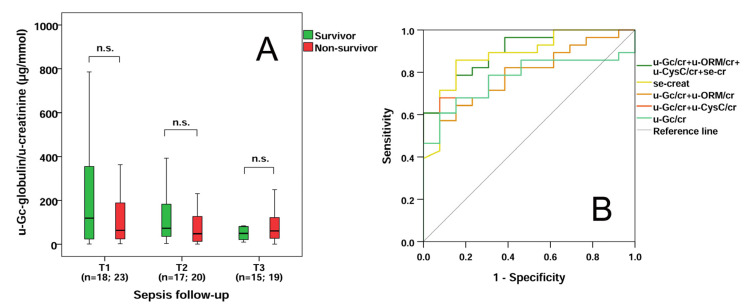
Survival and predictive power of u-Gc/u-creatinine alone and in combination with other markers. U-Gc-globulin/u-creatinine levels in survivor and non-survivor septic patients based on 14-day mortality during follow-up (**A**). Receiver operating characteristic (ROC) curves of admission laboratory parameters alone and in various combinations of these markers for diagnosing sepsis-related AKI (**B**). n: sample count, n.s.: non-significant.

**Figure 5 molecules-28-06864-f005:**
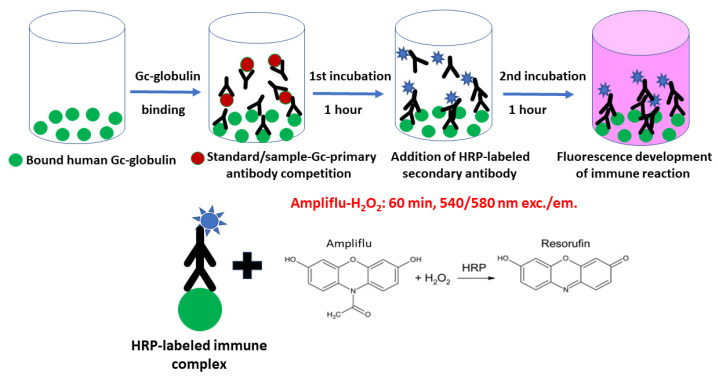
Flow chart and reaction scheme of the fluorescence u-Gc Elisa method.

**Table 1 molecules-28-06864-t001:** Imprecision data for our u-Gc assay at two concentration levels of diluted Human Serum Protein Low Control (DAKO-Agilent). For evaluation of the inter-assay imprecision 10 independent measurements with 8 replicates at each concentration were used.

Imprecision Data for the u-Gc Method			
**u-Gc Intra-assay Imprecision**			
u-Gc target ng/mL (n = 25)	u-Gc obtained ng/mL (n = 25)	SD	CV%
7.81	8.08	1.08	13.33
31.25	31.16	2.55	8.20
**u-Gc Inter-assay Imprecision**			
u-Gc target ng/mL (n = 80)	u-Gc obtained ng/mL (n = 80)	SD	CV%
7.81	7.60	1.54	20.30
31.25	31.29	3.82	12.21

**Table 2 molecules-28-06864-t002:** Patients’ demographic and admission laboratory data.

	Control (n = 23)	Sepsis (n = 13)	Sepsis + AKI (n = 28)	*p* Value
Age (years)	52 (48–56)	72 (58–80)	67 (59–70)	<0.05 ^a,b^
Males, n (%)	13 (56.5)	7 (53.8)	20 (71.4)	n.s.
**Major Comorbidities, n (%)**				
Cardiovascular disease	10 (43.5)	12 (92.3)	21 (75.0)	<0.05 ^a,b^
Type-2 diabetes mellitus	5 (21.7)	3 (23.1)	8 (28.6)	n.s.
Chronic kidney disease	0	3 (23.1)	4 (14.3)	<0.05 ^a^
Pulmonary disease	2 (8.7)	1 (7.7)	6 (21.4)	n.s.
Immunological disease	1 (4.3)	1 (7.7)	1 (3.6)	n.s.
Malignancy	0	3 (23.1)	10 (35.7)	<0.05 ^a,b^
**Admission Laboratory Data**				
se-TP (g/L)	76.1 (72.2–77.7)	41.6 (38.6–46.9)	48.7 (42.5–51.1)	<0.05 ^a,b,c^
se-albumin (g/L)	49.2 (46.9–51.1)	22.2 (18.9–26.4)	23.6 (19.6–28.4)	<0.05 ^a,b^
se-creatinine (µmol/L)	76 (70–86)	87 (64–136)	181 (146–297)	<0.05 ^b,c^
WBC (G/L)	7.2 (6.4–7.9)	13.8 (9.8–20.5)	17.1 (12.6–20.6)	<0.05 ^a,b^
PLT (G/L)	262 (249–300)	223 (133–306)	218 (140–321)	n.s.
hs-CRP (mg/L)	1.3 (0.6–2.5)	295.4 (189–367)	281.3 (167–390)	<0.05 ^a,b^
PCT (ng/mL)	-	7.9 (2.4–23.4)	11.6 (6.2–60.4)	n.s.
se-GSN (mg/L)	78.5 (75.1–89.1)	11.2 (6.5–25.2)	14.5 (6.1–21.1)	<0.05 ^a,b^
se-Gc (mg/L)	403.2 (369–418)	221.2 (205–251)	239.6 (178–279)	<0.05 ^a,b^
u-TP (mg/L)	60 (50–80)	120 (40–425)	270 (145–478)	<0.05 ^b^
u-albumin (mg/L)	4.7 (2.0–7.1)	8.1 (1.9–21.2)	38.8 (26.3–87.6)	<0.05 ^b,c^
u-ORM (mg/L)	0.9 (0.5–2.1)	31.9 (13.5–62.7)	31.5 (21.6–84.4)	<0.05 ^a,b^
u-ORM/u-creatinine (mg/mmol)	0.1 (0.05–0.17)	15.1 (10.9–22.2)	16.7 (11.3–24.2)	<0.05 ^a,b^
u-Cystatin C (mg/L)	0.06 (0.05–0.1)	0.53 (0.1–1.1)	1.17 (0.2–5.5)	<0.05 ^a,b^
u-Cystatin C/u-creatinine (mg/mmol)	0.006 (0.005–0.01)	0.19 (0.04–0.43)	0.39 (0.09–3.17)	<0.05 ^a,b^
u-Gc (ng/mL)	4.7 (0.1–32.5)	61.3 (22–257)	349.8 (106–1041)	<0.05 ^a,b,c^
u-Gc/u-creatinine (µg/mmol)	0.5 (0.1–2.8)	23.6 (14.4–63.1)	136.5 (39.9–379.1)	<0.05 ^a,b,c^

Continuous variables are shown as median (25th–75th percentiles) and categorical variables are expressed as a number (percentage). Mann–Whitney U and Chi-square tests were used for data comparison between patient groups. Level of significance was set at *p* < 0.05; n.s.: non-significant. Superscript lowercase letters refer to post hoc analyses: a: *p* < 0.05 between Control and Sepsis; b: *p* < 0.05 between Control and Sepsis + AKI; c: *p* < 0.05 between Sepsis and Sepsis + AKI groups.

**Table 3 molecules-28-06864-t003:** Clinical and microbiological data of septic patients.

	Sepsis (n = 13)	Sepsis + AKI (n = 28)	*p* Value
Age (years)	72 (58–80)	67 (59–70)	n.s.
Males, n (%)	7 (53.8)	20 (71.4)	n.s.
**Cause of Admission**			
Internal medicine origin, n (%)	2 (15.4)	12 (42.9)	n.s.
Surgical origin, n (%)	11 (84.6)	16 (57.1)	n.s.
ICU treatment days	9 (3–17)	11 (4–17)	n.s.
14-day mortality, death (%)	5 (38.5)	13 (46.4)	n.s.
AKI requiring RRT, n (%)	-	12 (42.9)	-
**Organ Dysfunctions, n (%)**			
1	4 (30.7)	3 (10.7)	n.s.
2	3 (23.1)	7 (25.0)	n.s.
≥3	6 (46.2)	18 (64.3)	n.s.
**Clinical Prognostic Scores**			
APACHE II score	15 (11–19)	23 (18–27)	<0.05
SAPS II score	42 (35–46)	53 (40–59)	<0.05
SOFA score	9 (7–11)	11 (9–13)	n.s.
**Identified Pathogens, n (%)**			
Unidentified	4 (30.7)	9 (32.1)	n.s.
Gram-positive bacteria	1 (7.7)	4 (14.3)	n.s.
Gram-negative bacteria	3 (23.2)	5 (17.9)	n.s.
Fungi	1 (7.7)	2 (7.1)	n.s.
Mixed	4 (30.7)	8 (28.6)	n.s.

Continuous variables are shown as median (25th–75th percentiles) and categorical variables are expressed as a number (percentage). Mann–Whitney U and Chi-square tests were performed for data comparison between patient groups. Level of significance is set at *p* < 0.05; n.s.: non-significant. RRT: renal replacement therapy.

**Table 4 molecules-28-06864-t004:** Significant correlations for u-Gc/u-creatinine vs. serum and urinary parameters.

U-Gc/u-Creatinine (µg/mmol)—Correlations (Spearman)	
**Parameter**	**Correlation Coefficient**
u-Gc-globulin (ng/mL)	0.923 (*p* < 0.001)
se-creatinine (µmol/L)	0.466 (*p* < 0.001)
se-urea (mmol/L)	0.399 (*p* < 0.001)
se-hs-CRP (mg/L)	0.407 (*p* < 0.001)
White Blood Cell count (G/L)	0.376 (*p* < 0.001)
se-total protein (g/L)	−0.402 (*p* < 0.001)
se-albumin (g/L)	−0.427 (*p* < 0.001)
se-gelsolin (mg/L)	−0.390 (*p* < 0.001)
se-Gc-globulin (mg/L)	−0.250 (*p* < 0.01)
u-total protein (mg/L)	0.563 (*p* < 0.001)
u-albumin (mg/L)	0.696 (*p* < 0.001)
u-ORM (mg/L)	0.415 (*p* < 0.001)
u-ORM/u-creatinine (mg/mmol)	0.608 (*p* < 0.001)
u-Cystatin C (mg/L)	0.624 (*p* < 0.001)
u-Cystatin C/u-creatinine (mg/mmol)	0.722 (*p* < 0.001)

## Data Availability

The data that support the findings of this study are available from the corresponding author upon reasonable request.

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
