# Peer review of "Measurement of Urinary Gc-Globulin by a Fluorescence ELISA Technique: Method Validation and Clinical Evaluation in Septic Patients—A Pilot Study"

_molecules, 2023, doi:10.3390/molecules28196864_

Round 1

Reviewer 1 Report

In this manuscript, the authors processed a pilot study of the measurement of urinary Gc-globulin by a fluorescence ELISA technique, which was used to validate and evaluate the Gc-globulin level in serum and urine samples from the septic patients. Despite providing a certain amount of data, the developed fluorescence ELISA technique has no innovation, and there lack a schematic description of the technique. Besides, there is no comparison between their technique and other methods, which is critical to demonstrate the advantage of their technique and the reliability of their data. Thus, the reviewer can’t support the publication of this manuscript.

Reviewer 2 Report

The manuscript ID: molecules-2614687 entitled “Measurement of Urinary Gc-globulin by a Fluorescence ELISA Technique: Method Validation and Clinical Evaluation in Septic Patients - a Pilot Study” by KÅ‘szegi et al. can be published in Molecules after minor revision.

Specific comments      

According to the Authors, a competitive fluorescence ELISA method for urinary Gc-globulin (u-Gc) measurement was developed and validated. Next, this method was applied for the quantification of u-Gc to  assess the potential application of this diagnostic marker for sepsis-induced acute kidney injury (AKI). Thus, serum and urine samples from septic patients were collected in 3 consecutive days (T1, T2, T3) and data were compared to controls. The obtained results showed that the increased general permeability in sepsis, especially in kidney injury, causes the urinary Gc-globulin levels. Therefore, urinary Gc-globulin seems to be a promising additional diagnostic marker for sepsis-induced acute kidney injury. Thus, u-Gc combined with other glomerular and tubular injury markers (se-creatinine, u-orosomucoid, u-Cystatin-C, u-actin) might give reliable clinical information regarding sepsis associated acute kidney injury.

The topic of the study are very interesting and important taking into the fact that sepsis is still a highly challenging and complex syndrome for intensive care patients worldwide with a mortality rate between 30 - 50%. The paper is well written and the statistical evaluation of the obtained results correctly performed. In my opinion, this paper can be published in Molecules after minor revision.

 2.3. Patients’ Demographic and Laboratory Data

The authors reported that “Admission values of se-ALB, se-creatinine, WBC, hsCRP, se-GSN, se-Gc, u-TP, u-ALB, u-ORM, u-ORM/u-creatinine, u-Cystatin C and u-Cystatin C/u-creatinine were also significantly different (p<0.05) in both septic groups compared with the corresponding data of the control patients.”. However, the results of statistical analysis presented in Table 2 for  se-creatinine, u-TP and u-albumin (u-ALB) are inconsistent with these data. It should be explained.

Summarizing, the paper after minor revision can be published in Molecules.

Round 2

Reviewer 1 Report

The authors have made significant improvements in the revised manuscript, now the reviewer support the publication of this manuscript.